# Novel Blood Collection Tubes Improve Sample Preservation in a Multicenter Study in Thailand

**DOI:** 10.3390/diagnostics15182398

**Published:** 2025-09-20

**Authors:** Khundow Moonla, Renu Wiriyaprasit, Napaporn Apiratmateekul, Nam K. Tran, Wanvisa Treebuphachatsakul

**Affiliations:** 1Reference Material and Innovation Research Unit, Faculty of Allied Health Sciences, Naresuan University, Phitsanulok 65000, Thailand; kwabdaom64@nu.ac.th (K.M.); napaporna@nu.ac.th (N.A.); 2Department of Medical Technology, Faculty of Allied Health Sciences, Naresuan University, Phitsanulok 65000, Thailand; renuw103@gmail.com; 3Computational Pathology and AI Center of Excellence (CPACE), School of Medicine, University of Pittsburgh, Pittsburgh, PA 15213, USA; trannk@upmc.edu

**Keywords:** analytes, clinical investigation, diabetes, triglycerides, thyroid hormones, glucose stability, glycolysis pathway, falsely low, routine biochemical tests, a single blood for essential testing

## Abstract

**Background**: Blood collection tubes (BCTs) are critical in vitro diagnostic devices used in clinical laboratory testing. Innomed tubes are novel BCTs coated with heparin and anti-glycolytic agents (Innomed 1) and clot activators combined with anti-glycolytic agents (Innomed 2). This study, we focus on the verification and multicenter validation of Innomed tubes, aiming to assess their performance in glucose stability, hemolysis resistance, and biochemical interferences relevant to diabetes and non-communicable disease (NCD) monitoring. **Methods**: Two types of Innomed tubes were evaluated. The verification process, conducted in a reference laboratory, assessed biochemical interferences, glucose stability, and the potential for hemolysis due to centrifugation and blood collection techniques. The multicenter validation of biochemical interferences was carried out across five hospital laboratories. Subsequently, samples were centrifuged at 3500 rpm for 5–10 min and analyzed immediately after separation, not exceeding 2 h from collection. **Results**: Glucose levels in Innomed 1 and 2 remained stable within 8% of up to 8 h post-collection. No hemolysis was observed under varying centrifugation times (at 3500 rpm) and blood collection techniques, as confirmed through visual inspection and lactate dehydrogenase level determinations. Innomed tubes were suitable BCTs for glucose, HbA1c, thyroid stimulating hormone, free triiodothyronine, free thyroxin, triiodothyronine, carcinoembryonic antigen, and prostate-specific antigen, as well as for 19 routine biochemical assays. **Conclusions**: Innomed 1 and 2 BCTs maintain blood glucose stability for 8 h, ensuring accurate biochemical, HbA1c, thyroid, and tumor marker testing. Their hemolysis resistance supports diabetes screening and Non-Communicable Diseases (NCD) monitoring. thereby emphasizing their clinical relevance in chronic disease management.

## 1. Introduction

Blood specimens are the most commonly used samples in laboratory tests. In Thailand, blood collection tubes (BCTs) are classified as in vitro diagnostic medical devices and must be manufactured in Food and Drug Administration (FDA)-certified facilities and adhere to the International Organization for Standardization (ISO) 13485 standards: Medical devices-quality management systems-requirements for regulatory purposes [1]. There are various types of BCTs, including plain tubes, those with anticoagulants, and tubes with clot activators, which are used in various tests. Multi-biochemical testing using a single blood sample is becoming increasingly important for using a single blood collection tube offers practicality, thriftiness, and improved patient care, as blood is collected in one step rather than multiple tubes, thereby reducing patient discomfort and pre-analytical variability [2,3,4]. Glucose testing is a common biochemical test, often using sodium fluoride (NaF) tubes. NaF acts as an anticoagulant and inhibits glycolysis. However, it exhibits delayed inhibition and a tendency to cause hemolysis [5]. To address these issues, NaF is often combined with other anticoagulants. For example, citrate-buffered NaF-ethylenediaminetetraacetic acid (EDTA) tubes, which represent an alternative, yield overestimated glucose levels, limiting their use [6]. Conversely, heparin tubes, commonly coated with lithium salt, allow for rapid plasma separation; however, they may cause hemolysis if not used correctly. Although heparin is suitable for routine biochemical tests, it does not inhibit glycolysis, leading to underestimated glucose levels if there is a delay in processing [7].

Hemolysis is a significant preanalytical error affecting approximately 3% of samples, rendering approximately 60% of those unsuitable for analysis. Hemolysis can be caused either by improper collection techniques or underlying medical conditions [8,9]. Blood collection is typically performed using one of the two main methods. First, the needle-syringe method allows for control over the blood volume and is suitable for small samples. Second, the vacuum blood collection system minimizes air bubbles and cell damage, accelerating extraction and reducing the risk of hemolysis [10].

BCTs require rigorous verification and validation to ensure safety and accuracy. Verification confirms that tubes meet the design requirements, whereas validation ensures reliable performance in clinical settings [11]. These processes are essential for maintaining sample integrity and ensuring patient safety [12]. Substandard BCTs can lead to sample degradation, whereas those with appropriate chemicals can preserve sample integrity, ensuring consistent and comparable results. Innomed tubes [13,14] are novel BCTs coated with heparin and anti-glycolysis agents (Innomed 1) and clot activators combined with anti-glycolysis agents (Innomed 2). Innomed tubes enable measurements of multiple biochemical parameters, including glucose, within the same tube and can preserve glucose concentration for up to 8 h after blood collection. These tubes are FDA-approved in Thailand, and their production processes are certified according to the international standard ISO 13485 [1]. The Innomed tubes have been optimized with precise coating proportions to suit their intended application, and rigorous validation and verification processes have been conducted to ensure their efficacy and reliability [4]. The clinical relevance of this study is underscored by the capacity of Innomed tubes to preserve glucose stability within clinically acceptable limits for up to 8 h and to resist hemolysis under different processing conditions, thereby supporting accurate biochemical testing and reliable monitoring of diabetes and other non-communicable diseases.

In the present study, we aimed to verify and validate the Innomed BCTs by comparing them to commercial tubes and ensuring that their use does not lead to hemolysis when subjected to two distinct centrifugation and blood collection techniques. The verification and validation were conducted according to the Clinical and Laboratory Standards Institute (CLSI) GP34A standards.

## 2. Material and Methods

### 2.1. Production of Innomed Tubes

This study was approved by the Institutional Review Board of Naresuan University (IRB No. P3-0093/2564, date of approval 9 August 2021). The Innomed tubes were manufactured by the WE Med Lab Center Co., Ltd. (Phitsanulok, Thailand). These vacuum tubes were coated with either heparin anticoagulants (Innomed 1) or clot activators (Innomed 2), along with anti-glycolysis agents in optimal amounts. Manufacturing equipment and procedures adhered to ISO 13485 standards. Additionally, their production met the requirements for post-production quality control and physical testing in accordance with ISO 6710 [15]. This includes tests for draw volume, leakage, and robustness, ensuring no leaks in the final product. The discrepancy between the volume indicated on the tube label and the actual amount of vacuum-aspirated blood was within 10%.

### 2.2. Verification of Innomed Tubes

#### 2.2.1. Biochemical Interference

The Innomed 1 and 2 BCTs were verified in accordance with CLSI GP34A: Validation and Verification of Tubes for Venous and Capillary Blood [11] by the Medical Device Research Laboratory at Naresuan University, Phitsanulok, Thailand, as per the ISO 15189 standards [16]. Innomed 1 and 2 tubes were used to collect venous blood samples from 36 participants, comprising 12 healthy individuals and 24 patients. For each donor, venous blood was collected in parallel into a commercial tubes (control) and an Innomed tube (test), enabling direct comparison between tube types. Healthy participants were aged ≥20 years, free from disease, had no recent injuries or blood loss, and laboratory test results within reference ranges. The patient group included 8 with diabetes mellitus, 8 with hypercholesterolemia, and 8 with multiple chronic non-communicable diseases (e.g., diabetes combined with hypercholesterolemia or hypertension), also aged ≥20 years and without recent injuries or blood loss. Both groups had equal numbers of male and female participants. Exclusion criteria included pregnancy, anemia, cancer, hepatitis B or C infection, immunodeficiency (HIV), syphilis, or inability to provide sufficient blood volume [11]. Biochemical and hemoglobin (Hb) A1c analyses were performed using the Cobas Integra 400 automated analyzer (Roche Diagnostics, Basel, Switzerland), and complete blood count (CBC) was analyzed using the ABX Pentra XL 80 analyzer (HORIBA ABX, Montpellier, France). The following parameters were measured: glucose; 19 routine biochemical analytes (albumin, ALP, ALT, AST, direct bilirubin, total bilirubin, BUN, cholesterol, creatinine, HDL, LDH, LDL, total protein, triglycerides, uric acid, sodium, potassium, chloride, bicarbonate); HbA1c; thyroid hormones (TSH, FT3, FT4, T3); tumor markers (CEA, PSA); and complete blood count parameters (hemoglobin, hematocrit, RBC, WBC, platelet count, MCV, MCH, MCHC, RDW, and differential counts of neutrophils, lymphocytes, monocytes, eosinophils, and basophils).

#### 2.2.2. Glucose Stability

Five production lots of Innomed 1 and four lots of Innomed 2 tubes were evaluated for glucose stability in accordance with the quality control process of the manufacturer. BCTs were randomly sampled from each lot, and the blood samples were collected into selected tubes. Three tubes were used for each lot to collect the plasma and serum samples. The glucose levels in these samples were measured and compared to those of NaF/K_3_EDTA tubes to assess the glucose stability of Innomed BCTs from baseline to 8 h. Innomed 1 and 2 tubes were verified for glucose stability using samples collected from 24 participants using the Cobas Integra 400 automated analyzer (Roche Diagnostics).

### 2.3. Hemolysis Investigation of Centrifugation Time and Blood Collection Techniques

To determine the impact of centrifugation time on measurement values, blood samples were collected from 10 healthy participants. The blood of each volunteer was placed into Innomed 1 and 2 BCTs. The samples were then centrifuged for either 5 or 10 min at 3500 rpm and subsequently analyzed to measure lactate dehydrogenase (LDH) levels using the Cobas Integra 400 automated analyzer (Roche Diagnostics).

To assess the effects of different blood collection techniques on analysis, blood samples were obtained from 20 healthy participants. For this assessment, the syringe technique was employed using a 21-gauge needle and a 10 mL syringe to collect blood from one arm of each volunteer. The other arm was used for blood collection via a vacuum blood collection system. These procedures were performed by a skilled medical technician. Once collected, the blood samples were analyzed for LDH levels using the Cobas Integra 400 automated analyzer (Roche Diagnostics).

### 2.4. Validation of Innomed Tubes

After passing the verification phase, the Innomed BCTs were validated in the clinical laboratories of five hospitals in a multicenter study using various biochemical tests (Appendix A). Additional major electrolytes and thyroid hormones were validated by the end users according to ISO 13485 standards at multicenter sites. The inclusion criteria for selecting these five clinical laboratories required accreditation of the laboratory at the hospital by ISO 15189 or the national standard of clinical laboratory by the Medical Technology Council of Thailand. Biochemical parameters of blood samples were measured in the five clinical laboratories within 2 h of collection.

### 2.5. Data Analyses

Plasma or serum glucose levels obtained from blood samples collected into Innomed 1 and 2 tubes were measured from the zero time point (baseline) to 8 h at room temperature (25 ± 2 °C) and relative humidity of 73 ± 3%. Bias was calculated as the percentage change from the baseline. Data distribution was assessed for normality using the Shapiro–Wilk test. Comparisons of glucose levels were performed using a paired *t*-test to compare the baseline values between Innomed and NaF/K_3_EDTA tubes, as well as the values for the Innomed tubes at the baseline and 8 h. Data were considered statistically significant at *p* < 0.05.

Comparisons between paired mean differences obtained in laboratory tests were performed using paired *t*-tests. Statistical significance was set at *p* < 0.05. Biased desirable quality specifications (Bias_d_) can have different effects on clinical decisions made based on the desirable biological variation database specifications recommended by Westgard [17]. The Johns Hopkins Hospital Laboratory Acceptability Thresholds (JHH) [18] criteria were used to establish the clinical limits for measurements, ensuring accurate laboratory results tailored to the practical needs of medical testing. This study prioritized the JHH threshold criteria as the primary acceptable clinical limits; however, the Bias_d_ was applied for certain tests when the JHH criteria were not available.

Non-parametric statistical methods were utilized in centrifugation tube analysis, given the small sample size (*n* = 10). The median and interquartile range (IQR) were calculated for LDH levels at 5 and 10 min centrifugation times, where the median indicates central tendency and the IQR assesses data spread. Normality of LDH level data was assessed using the Shapiro–Wilk Test prior to selecting the appropriate statistical method. The Mann–Whitney U test was applied to evaluate the hypothesis that centrifugation times do not significantly impact LDH levels. Statistical significance was set at *p* < 0.05. Mean, median, IQR, and bias were used to compare blood collection techniques. While the mean and median provided measures of LDH level centrality, the IQR quantified variability. Bias in percentage (%) was calculated as follows:Bias (%) = (mean experimental − mean reference)/mean reference × 100

Statistical significance was determined either using paired *t*-tests or the Mann–Whitney U test if the data were non-normally distributed. Statistical significance was considered at *p* < 0.05. A bias_d_ of 4.3% was used to evaluate LDH bias.

## 3. Results

### 3.1. Verification of Interference with Biochemical Tests

Innomed 1 and Innomed 2 exhibited no significant differences (*p* > 0.05) in the paired mean differences of 18 and 16 biochemical tests, respectively, compared to commercial values. The biases for these tests were within the accepted Biased and JHH criteria, when applicable, except for the LDH bias in Innomed 2, which exceeded the Bias_d_ threshold (Table 1). Additionally, Innomed 1 demonstrated no significant differences (*p* > 0.05) in the paired mean differences for HbA1c and 13 CBC parameters, except for the platelet count (Table 2), which displayed a bias exceeding the Bias_d_ criteria when compared to results from the commercial EDTA tube.

### 3.2. Verification of Glucose Stability

Five production lots of Innomed 1 and four lots of Innomed 2 maintained glucose levels within acceptable limits up to 8 h post-collection (Table 3), with no significant differences (*p* > 0.05) compared with those obtained from NaF/K_3_EDTA tubes. Glucose levels at the baseline (0 h) for NaF/K_3_EDTA tubes were not significantly different from those of the Innomed tubes. Moreover, the reduction in glucose at 8 h was no more than 8% when compared with the baseline level (0 h) for both Innomed 1 and 2 BCTs.

### 3.3. Verification of Hemolysis Assessment

#### 3.3.1. Centrifugation Time

Herein, no statistically significant differences were observed (*p* > 0.05), and biases in LDH levels did not exceed 4.3% when comparing the 5 and 10 min centrifugation times for Innomed 1 and 2 (Table 4).

#### 3.3.2. Blood Collection Techniques

The impact of needle-syringe and vacuum blood collection system on LDH levels was minimal and within 3.0% for Innomed 1 and 2 (Table 4). When compared with commercial tubes using the same blood collection techniques, the biases ranged from 1.0% to 3.0% for Innomed 1 and from 1.1% to 4.3% for Innomed 2. However, the differences were not significant for all comparisons (*p* > 0.05).

#### 3.3.3. Multicenter Validation of Interference on Biochemical Tests

The list of biochemical tests and characteristics of automated analyzers used for validation at five clinical laboratories is presented in Appendix A. The results of the comparison between Innomed 1 and commercial heparin tubes and Innomed 2 and commercial clot-activator tubes are presented in Table 5. Both Innomed 1 and 2 tubes exhibited no significant differences (*p* > 0.05) in paired mean differences when compared to commercial tubes. All biochemical tests adhered to the JHH and biased criteria.

## 4. Discussion

The stability of glucose in blood samples is a critical factor in ensuring accurate biochemical testing. Innomed tubes are novel BCTs coated with heparin and anti-glycolytic agents (Innomed 1) and clot activators combined with anti-glycolytic agents (Innomed 2). The present study demonstrated that the Innomed 1 and Innomed 2 BCTs can effectively inhibit the glycolysis pathway [4], preserving glucose levels. Innomed tubes significantly reduced the rate of glucose degradation, maintaining glucose stability within 8% of initial levels up to 8 h post-collection. This extended stability is crucial in clinical settings where immediate processing of blood samples may not always be feasible, ensuring that glucose levels remain reliable for diagnostic purposes. The findings of this study align with previous research that highlights the importance of using anti-glycolytic agents to inhibit glycolysis and preserve glucose levels for accurate testing [4]. In this study, glucose stability was assessed for up to 8 h, reflecting the routine preanalytical timeframe in hospital laboratories and consistent with CLSI GP34A recommendations. We acknowledge that in some settings samples may be stored longer (e.g., 24 h), and further studies under extended conditions would be valuable.

Delayed sample processing is a frequent preanalytical limitation in outpatient and resource-constrained settings and can artifactually depress plasma glucose, leading to misclassification near diagnostic thresholds and undermining longitudinal monitoring [5,7,9]. In our study, Innomed tubes preserved glucose within ±8% for up to 8 h post-collection, thereby reducing the risk of diagnostically meaningful underestimation when immediate centrifugation or analysis is not feasible. By stabilizing glucose without hemolysis-related interference [19], these tubes may reduce repeat phlebotomy, streamline workflows, and support accurate case finding and follow-up in decentralized care. The observed stability window aligns with our verification/validation framework and clinically acceptable limits, supporting adoption in routine practice where transport delays are common.

In the present study, Innomed BCTs for biochemical tests exhibited minimal interference, with most parameters meeting the acceptable criteria set by the JHH. However, our results indicated that the Innomed tubes are not suitable for CBC analysis, particularly concerning platelet count. The Innomed 1 tubes exhibited significant bias in platelet counts, with a tendency for platelet clumping, leading to values below those obtained for commercial EDTA tubes [20]. As platelet aggregation could interfere with accurate CBC results, the Innomed tubes are unsuitable for this specific test. Our findings emphasize the importance of selecting appropriate BCTs for specific laboratory analyses to avoid erroneous results [21].

Hemolysis is a common preanalytical error that significantly affects the accuracy of biochemical tests [19]. Notably, we investigated the potential for hemolysis when using Innomed BCTs under various centrifugation times and blood collection techniques. Our results demonstrated no significant differences in LDH levels between samples centrifuged for 5 or 10 min, nor between those collected using needle-syringe versus vacuum blood collection system [22]. This suggests that the Innomed tubes are robust against hemolysis, regardless of the processing method used. The ability to maintain sample integrity without inducing hemolysis enhances the reliability of test results, making these tubes a reliable choice in clinical practice where hemolysis is a concern.

Our multicenter validation conducted across five clinical laboratories provided strong evidence for the reliability and consistency of the Innomed BCTs in routine biochemical testing. Our results revealed no significant differences for paired mean differences across the range of biochemical tests [23], including major electrolytes and thyroid hormones, when compared against commercial tubes. This consistency across diverse clinical environments underscores the robustness of the Innomed tubes in maintaining sample integrity and ensuring accurate test results. Successful validation in multiple settings supports the utilization of these tubes in various healthcare environments, offering confidence in their performance across different laboratory conditions.

Notably, the production of Innomed BCTs was compliant with the ISO 13485 standards, which are internationally recognized guidelines for the manufacturing of medical devices. These standards ensure that tubes are produced with a high degree of quality control, covering aspects such as draw volume, leakage, and robustness. The adherence to ISO 13485 not only guarantees the safety and reliability of the Innomed tubes but also facilitates their acceptance in both domestic and international markets. The ability to produce these tubes at an industrial scale, while maintaining stringent quality standards, positions Innomed BCTs as a viable option for widespread clinical use, supporting the advancement of healthcare diagnostics. Moreover, the use of Innomed BCTs offers several cost-effective advantages in clinical practice. The extended glucose stability of these tubes allows for flexible sample processing times, reducing the urgency of immediate analysis and potentially lowering the laboratory operating costs. Additionally, the ability to perform multiple tests on a single sample with minimal interference improves workflow efficiency and saves time. Our data also suggest that the use of Innomed tubes can reduce the amount of generated infectious waste by approximately 20%, as fewer tubes are required for comprehensive testing. This waste reduction lowers disposal costs and contributes to environmental sustainability in healthcare and is aligned with global efforts to minimize the environmental footprint of medical practices [24].

This study has certain limitations. Blood glucose measurement primarily employs enzymatic methods such as hexokinase, glucose oxidase, and oxidation-reduction based on wet and dry chemistry. However, our study focused on the hexokinase method with wet chemistry, given its robust performance against interfering substances; therefore, data on other enzymes and methods are lacking. Future studies need to focus on screening infectious diseases and cardiac markers to provide benefits to patients in emergency settings. This study did not specifically include or stratify participants with leukocytosis (elevated white blood cell counts). Because higher white blood cell counts can accelerate in vitro glycolysis and spuriously lower measured plasma glucose concentrations [25], further studies are needed.

Overall, the findings of this study that the Innomed 1 and Innomed 2 BCTs are highly effective in clinical use and are suitable for measuring glucose levels, HbA_1c_, thyroid hormones (such as thyroid stimulating hormone, free triiodothyronine, free thyroxine, and triiodothyronine), and tumor markers, including prostate-specific antigen and carcinoembryonic antigen. They are also suitable for 19 routine biochemical tests, namely albumin, alkaline phosphatase, alanine aminotransferase, aspartate aminotransferase, direct bilirubin, total bilirubin, blood urea nitrogen, cholesterol, creatinine, high-density lipoprotein, LDH, low-density lipoprotein, total protein, triglycerides, uric acid, Na^+^, K^+^, Cl^−^, and HCO_3_^−^. Moreover, these tubes could maintain stable blood glucose levels for up to 8 h after collection, making them ideal for use in wellness programs and for patients with non-communicable diseases who require frequent biochemical blood testing. Notably, the Innomed tubes demonstrated robustness against hemolysis with different centrifugation times. Specifically, they can be effective within 5 to 10 min, offering flexibility in sample processing times without compromising sample integrity, regardless of the blood collection techniques used. This reliability ensures that the Innomed tubes maintain sample integrity, making them suitable for routine clinical practice.

## 5. Conclusions

The Innomed 1 and Innomed 2 BCTs are highly effective in clinical use and are suitable for measuring glucose levels, 19 routine biochemical tests, hemoglobin A1c, thyroid hormones, and tumor markers. These tubes maintain stable blood glucose levels for up to 8 h after collection, making them ideal for use in wellness programs. In this study, the wellness program is preventive health services and NCD monitoring supported by reliable laboratory testing with Innomed tubes. Such integration may enhance preventive care and chronic disease management within healthcare systems and for patients with non-communicable diseases who require frequent biochemical blood testing. Additionally, Innomed tubes are robust against hemolysis when different centrifugation times and blood collection techniques are used. This reliability ensures that the tubes maintain sample integrity, making them suitable for routine clinical practice. This study encompassed clinical and multicenter evaluations in Thailand, confirming the applicability of Innomed tubes. Further investigations in larger and more diverse populations and healthcare systems are warranted to enhance generalizability.

## Figures and Tables

**Table 1 diagnostics-15-02398-t001:** Pre-clinical evaluation of Innomed 1 and Innomed 2 for biochemical analysis compared to the commercial tube in healthy group (*n* = 12) and patient group (*n* = 24).

**Parameter**	**Clinical** **Significance Limits**		**Healthy**						**Patient**					
			**Bias (%)**	**Heparin**		**Innomed 1**		** *p* ** **-Value**	**Bias (%)**	**Heparin**		**Innomed 1**		** *p* ** **-Value**
	**JHH** **(%)**	**Bias_d_ (%)**		**Range**	**Mean (SEM)**	**Range**	**Mean (SEM)**			**Range**	**Mean (SEM)**	**Range**	**Mean (SEM)**	
ALB (g/L)	7	1.43	0.00	41.10–48.1	45 (0.7)	41.10–47.8	45 (0.6)	0.786	0.00	37–45	43 (1.1)	39.3–67.00	43 (1.1)	0.493
ALP (U/L)	10	6.72	0.69	38.90–75.70	57.60 (3.46)	38.47–75.80	57.20 (3.35)	0.964	1.81	42.33–136.33	75.80 (4.54)	45.00–144.33	75.80 (4.54)	0.826
ALT (U/L)	15	11.48	0.56	12.67–28.93	17.90 (1.79)	11.83–29.60	18.00 (1.92)	0.964	5.73	4.00–34.00	16.60 (1.58)	4.33–35.00	16.60 (1.58)	0.695
AST (U/L)	10	6.54	1.61	6.20–24.20	12.40 (1.91)	6.33–24.30	12.60 (1.92)	0.937	2.22	10.67–36.33	23.00 (1.32)	11.00–38.00	23.00 (1.32)	0.782
BIL-D (µmol/L)	NA	14.20	0.00	1.197–3.591	1.71 (0.17)	0.06–0.20	1.02 (0.17)	0.756	0.00	1.37–5.30	3.42 (0.17)	1.03–4.28	3.42 (0.17)	0.371
BIL-T (µmol/L)	10	8.95	0.00	2.05–2.05	6.84 (1.37)	1.71–18.47	6.84 (1.37)	0.865	8.18	1.88–12.83	6.50 (0.68)	1.20–13.00	3.50 (0.68)	0.524
BUN (mmol/L)	10	5.57	0.89	2.63–5.15	4.00 (0.18)	2.70–5.33	4.03 (0.18)	0.858	2.48	2.86–6.07	4.21 (0.20)	2.86–6.06	4.21 (0.20)	0.689
CHOL (mmol/L)	5	4.10	0.89	3.01–7.72	4.66 (0.37)	3.20–7.59	4.63 (0.35)	0.934	3.10	4.06–7.025	5.02 (0.11)	4.20–6.73	5.02 (0.12)	0.399
CRE (µmol/L)	7	3.96	0.00	0.01–0.03	0.02 (0.00)	0.01–0.03	0.07 (0.00)	0.996	0.00	0.02–0.03	0.02 (0.00)	0.01–0.03	0.02 (0.00)	0.499
HDL (mmol/L)	10	5.61	1.46	1.16–2.34	1.60 (0.12)	1.12–2.25	1.58 (0.12)	0.882	0.64	0.80–1.89	1.20 (0.06)	0.82–1.87	1.20 (0.06)	0.911
LDH (U/L)	10	4.30	2.00	101.87–220.10	155.00 (9.81)	101.20–181.67	158.10 (7.45)	0.799	3.89	113.40–233.67	187.10 (7.18)	123.30–237.93	187.10 (7.18)	0.482
LDL (mmol/L)	10	5.46	0.68	1.22–5.94	2.68 (0.34)	1.33–5.80	2.66 (0.33)	0.970	3.79	2.27–5.27	3.16 (0.12)	2.01–4.84	3.16 (0.12)	0.790
TP (g/dL)	5	1.36	0.00	6.53–7.66	7.20 (0.12)	6.53–7.66	7.20 (0.10)	0.503	0.00	6.33–7.23	6.80 (0.05)	6.30–7.33	6.80 (0.05)	0.596
TRIG (mmol/L)	5	9.57	0.00	0.37–1.86	0.87 (0.15)	0.37–1.86	0.87 (0.14)	0.99	1.40	0.62–7.48	1.69 (0.29)	0.64–7.07	1.67 (0.27)	0.952
UA (µmol/L)	10	4.87	2.33	174.24–319.62	255.85 (13.69)	172.55–309.40	249.90 (12.50)	0.840	1.59	142.80–521.32	374.85 (19.04)	142.80–513.99	368.90 (12.50)	0.815
TSH (mIU/ML)		14.14	0.00	0.65–4.84	1.90 (0.30)	0.65–4.76	1.90 (0.29)	0.955	8.00	0.67–5.16	2.50 (1.50)	0.73–5.27	2.40 (1.60)	0.951
CEA (µg/L)		14.30	0.00	0.01–0.02	0.02 (0.02)	0.01–0.02	0.02 (0.01)	0.881	0.00	0.01–0.06	0.03 (0.00)	0.01–0.07	0.03 (0.00)	0.937
**Parameter**	**Clinical** **Significance Limits**		**Healthy**						**Patient**					
			**Bias (%)**	**Clot Activator**		**Innomed 2**		** *p* ** **-Value**	**Bias (%)**	**Clot Activator**		**Innomed 2**		** *p* ** **-Value**
	**JHH** **(%)**	**Bias_d_ (%)**		**Range**	**Mean (SEM)**	**Range**	**Mean (SEM)**			**Range**	**Mean (SEM)**	**Range**	**Mean (SEM)**	
ALB (g/L)	7	1.43	0.00	41.0–55.0	48.0 (1.30)	41.0–53.7	48.0 (1.20)	0.706	0.00	40.0–49.0	45.0 (0.80)	39.7–48.7	45.0 (0.60)	0.899
ALP (U/L)	10	6.72	0.92	41.00–5.00	76.50 (6.73)	40.33–104.00	75.80 (6.59)	0.942	6.21	41.00–148.33	80.50 (7.23)	40.33–150.33	80.50 (7.23)	0.658
ALT (U/L)	15	11.48	1.59	9.67–16.00	12.6 (0.73)	9.67–17.00	12.80 (0.84)	0.851	4.15	7.67–64.00	21.70 (3.01)	8.00–64.00	22.60 (2.93)	0.838
AST (U/L)	10	6.54	6.01	11.00–27.67	18.30 (1.27)	13.00–29.00	19.40 (1.20)	0.511	3.07	15.67–54.00	26.10 (2.18)	16.00–55.00	26.90 (2.21)	0.806
BIL-D (µmol/L)	NA	14.20	0.00	1.71–5.30	3.42 (0.34)	1.71–5.30	3.42 (0.34)	0.953	0.00	1.71–3.76	3.42 (0.17)	1.54–3.59	3.42 (0.17)	0.682
BIL-T (µmol/L)	10	8.95	0.00	3.93–13.00	6.84 (0.86)	4.10–13.00	6.84 (0.86)	0.814	0.00	3.25–9.75	6.84 (0.51)	4.10–10.43	6.84 (0.51)	0.511
BUN (mmol/L)	10	5.57	0.83	2.86–6.43	4.32 (0.32)	2.74–6.43	4.28 (0.32)	0.920	1.38	1.79–7.50	5.18 (0.33)	1.79–7.50	5.25 (0.34)	0.920
CHOL (mmol/L)	5	4.10	0.44	3.76–6.32	4.69 (0.25)	3.71–6.28	4.67 (0.25)	0.953	0.17	4.06–11.06	6.11 (0.32)	4.04–11.08	6.12 (0.32)	0.980
CRE (µmol/L)	7	3.96	0.00	53.04–101.66	70.72 (4.42)	53.04–99.99	70.72 (4.42)	0.868	0.00	51.27–129.95	79.56 (5.30)	52.18–130.82	79.56 (5.30)	0.995
HDL (mmol/L)	10	5.61	0.77	0.78–2.00	1.35 (0.11)	0.78–1.94	1.34 (0.11)	0.890	1.51	0.78–2.13	1.20 (0.09)	0.78–2.13	1.22 (0.09)	0.878
LDH (U/L)	10	4.30	10.79	127.33–212.83	164.10 (6.75)	160.20–233.77	164.10 (6.75)	0.073	8.26	137.03–254.94	174.30 (7.91)	147.93–292.17	188.70 (9.18)	0.242
LDL (mmol/L)	10	5.46	0.09	1.62–3.84	2.73 (0.22)	1.59–3.84	2.74 (0.22)	0.998	0.07	1.74–9.20	3.75 (0.36)	1.68–9.22	3.75 (0.35)	0.993
TP (g/L)	5	1.36	0.00	65.3–81.0	74.0 (1.60)	66.0–81.0	74.0 (1.50)	0.980	0.00	65.3–77.0	72.0 (0.80)	66.0–77.0	72.0 (0.70)	0.928
TRIG (mmol/L)	5	9.57	0.29	0.48–1.90	1.19 (0.11)	0.47–1.86	1.18 (0.11)	0.929	0.74	1.34–7.02	2.59 (0.34)	1.33–6.97	2.61 (0.32)	0.967
UA (µmol/L)	10	4.87	0.00	196.35–523.60	339.15 (2.98)	198.14–519.04	339.15 (29.75)	0.942	0.00	196.35–640.32	398.65 (28.56)	198.14–660.45	398.65 (29.16)	0.944
TSH (mIU/ML)	-	14.14	-	-	-	-	-	-	0.00	0.35–3.31	1.80 (0.30)	0.35–3.28	1.80 (0.29)	0.987
CEA (µg/L)	-	14.30	-	-	-	-	-	-	0.00	0.02–0.05	0.03 (0.00)	0.02–0.05	0.03 (0.00)	0.959

Note: Bias_d_ = desirable quality specifications for bias, SEM = standard error of the mean, Bias = difference between the compared tube results, BUN = Blood urea nitrogen, CRE = Creatinine, CHOL = Total cholesterol, TRIG = Triglyceride, HDL = High-density lipoprotein, LDL = Low-density lipoprotein, UA = Uric acid, AST = Aspartate aminotransferase, ALT = Alanine aminotransferase, ALP = Alkaline phosphatase, BIL-D = Direct bilirubin, BIL-T = Total bilirubin, ALB = Albumin, LDH = lactate dehydrogenase, %Bias: (Mean test − Mean target)/Mean target × 100, %Bias_d_: Bias desirable quality specifications for bias, %JHH: Percent changes in analyte (Johns Hopkins Hospital Laboratory Acceptability Thresholds) Acceptable criteria and significant limits of clinical limits and/or statistical limits (*p* > 0.05), NA: Not applicable.

**Table 2 diagnostics-15-02398-t002:** Pre-clinical evaluation of Innomed 1 for CBC and HbA1c compared to the commercial tubes.

Parameter	Clinical Significance Limits	Healthy (*n* = 12)						Patient (*n* = 24)					*p*-Value
		Bias (%)	EDTA		Innomed 1		*p*-Value	Bias (%)	EDTA		Innomed 1		
	Bias_d_ (%)		Range	Mean (SEM)	Range	Mean (SEM)			Range	Mean (SEM)	Range	Mean (SEM)	
HbA1c (%)	1.50	0.00	4.97–5.40	5.20 (0.04)	4.90–5.40	5.20 (0.05)	0.820	0.00	4.97–7.80	5.70 (0.11)	5.00–7.73	5.70 (0.11)	0.911
RBC (10^12^/L)	1.70	0.00	3.70–5.33	4.60 (0.15)	3.80–5.26	4.60 (0.15)	0.887	0.00	3.70–6.20	4.60 (0.13)	3.77–5.73	4.60 (0.12)	0.224
HGB (g/L)	1.84	1.68	71–148	119 (6.6)	72.7–146.3	121 (6.3)	0.911	1.52	105.3–152	132 (2.9)	105.70–151.7	130 (2.9)	0.123
HCT (L/L)	1.74	0.27	0.25–0.45	0.37 (0.02)	0.25–0.44	0.37 (0.02)	0.950	1.53	0.33–0.45	0.39 (0.008)	0.32–0.45	0.385 (0.008)	0.083
MCV (fL)	1.26	0.00	57.00–90.00	80.70 (3.28)	56.67–90.00	80.70 (3.32)	1.000	0.11	68.33–100.00	87.10 (1.69)	68.00–99.33	87.00 (1.70)	0.655
MCH (pg)	1.35	0.38	15.83–29.73	26.00 (1.35)	16.20–29.77	26.10 (1.35)	0.955	0.34	21.53–34.30	29.00 (0.65)	21.47–34.20	29.10 (0.64)	0.361
MCHC (g/L)	0.04	0.31	317.7–333	325 (1.9)	318.3–334	326 (2.3)	0.794	0.00	315–342	334 (1.4)	909.3–345.7	334 (1.8)	0.948
RDW (%)	1.70	0.72	11.30–18.63	13.90 (0.63)	11.47–18.80	14.00 (0.63)	0.920	0.00	10.20–17.83	12.80 (0.32)	10.07–17.47	12.80 (0.31)	0.958
PLT (10^9^/L)	5.90	41.33	184.3–383.67	264.00 (25.69)	41.33–332.67	154.90 (29.64)	0.005	12.34	152.3–304.00	218.80 (10.75)	106.00–287.67	191.80 (12.47)	0.010
WBC (10^9^/L)	6.05	5.66	3.60–10.60	5.30 (0.64)	3.60–9.77	5.00 (0.57)	0.785	5.08	3.67–8.40	5.90 (0.27)	3.93–8.10	5.60 (0.23)	0.389
NEU (%)	9.25	4.40	4.40–23.77	34.10 (2.08)	23.13–43.13	32.60 (2.52)	0.660	7.59	35.33–77.00	47.40 (1.86)	30.33–64.67	43.80 (1.72)	0.169
LYM (%)	5.90	0.32	0.32–55.60	63.10 (2.19)	50.17–75.17	62.90 (2.75)	0.949	4.77	27.33–54.00	44.00 (1.41)	30.00–60.00	46.10 (1.60)	0.798
MON (%)	13.20	12.50	12.50–0.57	1.60 (0.24)	0.47–3.10	1.80 (0.24)	0.631	10.42	2.67–8.00	4.80 (0.31)	1.67–8.33	4.30 (0.31)	0.287
EOS (%)	19.80	12.50	12.50–0.80	1.60 (0.41)	0.30–4.33	1.80 (0.45)	0.284	4.76	1.00–11.33	4.20 (0.59)	1.00–11.67	4.40 (0.55)	0.837
BAS (%)	15.40	0.00	0.00–0.33	0.60 (0.07)	0.37–0.83	0.60 (0.05)	0.883	11.11	0.33–1.00	0.90 (0.03)	0.67–1.33	1.00 (0.03)	0.702

Note: Bias_d_ = desirable quality specifications for bias, SEM = standard error of the mean, Bias = difference between the compared tube results, HbA1c = Hemoglobin A1C, WBC = White Blood Cell, RBC = Red Blood Cell, HCT = Hematocrit, MCV = Mean Corpuscular Volume, MCH = Mean Corpuscular Width, PLT = Platelet, NEU = Neutrophil, LYM = Lymphocyte, MON = Monocyte, EOS = Eosinophil, BAS = Basophil, %Bias: (Mean test − Mean target)/Mean target × 100, %Bias_d_: Bias desirable quality specifications for bias, Acceptable criteria and significant limits of clinical limits and/or statistical limits (*p* > 0.05).

**Table 3 diagnostics-15-02398-t003:** Verification glucose stability at 0 and 8 h post-collection in Innomed 1, Innomed 2, and NaF/K3EDTA blood collection tubes (*n* = 24).

*Singer Lot*																	
**Test/h** **Glucose** **(mmol/L)**		**NaF/EDTA**						**Innomed 1**						**Clinical** **Significance Limits**			**Statistical Limit**
		**Range**			**Mean**	**SEM**	**% Decreased**	**Range**			**Mean**	**SEM**	**%** **Decreased**	**% Bias**	**% Bias_d_**	**JHH**	***p*-Value**
0 h		3.86–7.87			4.83	0.14	0.00	4.17–7.50			4.88	0.12	0.00	1.03	2.34	8	0.742
8 h		3.56–7.71			4.74	0.15	1.84	3.91–6.89			4.51	0.11	7.62	4.92	2.34	8	0.190
**Glucose** **(mmol/L)**		**NaF/EDTA**						**Innomed 2**						**Clinical** **Significance Limits**			**Statistical Limit**
		**Range**			**Mean**	**SEM**	**% Decreased**	**Range**			**Mean**	**SEM**	**%** **Decreased**	**% Bias**	**% Bias_d_**	**JHH**	** *p* ** **-Value**
0 h		3.83–13.67			6.61	0.79	0.0	3.89–12.89			6.56	0.75	0.0	−1.09	2.34	8	0.947
8 h		3.56–13.28			6.38	0.80	3.36	3.44–12.72			6.17	0.77	5.9	−3.74	2.34	8	0.833
*Multiple-lot*																	
**Lot**	**Glucose**	**Innomed Tube 1**						**NaFK_3_EDTA Tube**						**Acceptable Criteria & Significance**			
														**Clinical Limits**		**Statistical Limits**	
	**(mmol/L)**	**Range**			**Mean**	**SEM**	**%** **Decreased**	**Range**			**Mean**	**SEM**	**%** **Decreased**	**% Bias**	**% Bias_d_**	**% JHH**	***p*-Value**
00-220919	0 h	4.55	–	4.61	4.60	0.00	0.0	4.56	–	4.56	4.59	0.00	0.0	0.24	2.34	8	0.423
	8 h	4.22	–	4.28	4.24	0.01	7.1	4.22	–	4.28	4.23	0.02	7.0	0.13			0.556
00-230518	0 h	11.73	–	11.91	11.82	0.02	0.0	11.84	–	11.92	11.75	0.03	0.0	0.56			0.422
	8 h	11.23	–	11.38	11.33	0.02	4.2	11.43	–	11.49	11.21	0.02	3.6	1.11			0.057
00-231109	0 h	10.11	–	10.17	10.14	0.02	0.0	10.11	–	10.11	10.11	0.00	0.0	0.27			0.182
	8 h	9.44	–	9.56	9.47	1.00	6.6	9.72	–	9.83	9.18	0.03	3.3	3.13			0.001
00-240129	0 h	8.67	–	8.72	8.65	0.06	0.0	8.83	–	8.83	8.52	0.00	0.0	1.57			0.002
	8 h	8.06	–	8.11	8.06	0.58	7.35	8.36	–	8.45	7.74	0.27	4.8	4.16			0.001
00-240503	0 h	9.11	–	9.20	9.14	0.03	0.0	9.18	–	9.21	9.09	0.01	0.0	0.54			0.128
	8 h	8.00	–	8.52	8.27	0.07	9.48	8.25	–	8.27	8.25	0.56	10.2	0.20			0.936
**Lot**	**Glucose**	**Innomed Tube 2**						**NaFK_3_EDTA Tube**						**Acceptable Criteria & Significance**			
														**Clinical Limits**		**Statistical Limits**	
	**(mmol/L)**	**Range**			**Mean**	**SEM**	**%** **Decreased**	**Range**			**Mean**	**SEM**	**% Decreased**	**% Bias**	**% Bias_d_**	**% JHH**	***p*-Value**
00-221121	0 h	11.73	–	11.91	11.83	0.02	0.0	11.84	–	11.92	11.89	0.03	0.0	0.56	2.34	8	0.422
	8 h	11.23	–	11.38	11.33	0.02	4.2	11.43	–	11.49	11.46	0.02	3.6	1.11			0.057
00-230626	0 h	8.22	–	8.33	8.28	0.03	0.0	7.89	–	8.00	7.94	0.06	0.0	4.20			0.016
	8 h	8.11	–	8.11	8.18	0.06	1.14	7.44	–	7.56	7.50	0.06	5.59	9.11			0.003
00-231109	0 h	10.11	–	10.17	10.14	0.02	0.0	10.11	–	10.11	10.11	0.00	0.0	0.27			0.182
	8 h	9.44	–	9.56	9.47	0.06	6.6	9.72	–	9.83	9.78	0.03	3.3	3.13			0.001
00-240209	0 h	11.33	–	11.44	11.38	0.01	0.0	11.50	–	11.56	11.53	0.03	0.0	1.30			0.027
	8 h	10.67	–	10.78	10.74	0.06	5.62	10.83	–	10.89	10.86	0.03	5.78	1.13			0.528

**Table 4 diagnostics-15-02398-t004:** Comparison of centrifugation times and blood collection techniques on LDH levels in different blood collection tubes. (*n* = 10).

Parameter Centrifugation Times		5 min Centrifugation *				10 min Centrifugation *					*p*-Value
		Median	IQR			Median	IQR				
LDH (U/L)	Innomed 1	174	131	-	229	174	129	-	228		0.859
	Innomed 2	169	128	-	217	169	130	-	216		0.915
LDH (U/L)	**Blood Collection Techniques**	**Blood Collected by Needle Syringe System**				**Blood Collected by Vacuum System**					**Comparison** **of Blood** **Collection** **Methods (*p*)**
		**Mean**	**Median (IQR)**	**% Bias**	** *p* ** **-Value**	**Mean**	**Median (IQR)**	**% Bias**	** *p* ** **-Value**	**Bias_d_**	
	Innomed 1	175	174 (131–223)	3.0	0.654	179	174 (131–229)	1.0	0.879	4.3	0.754
	Heparin tube	170	172 (128–195)			177	172 (117–172)				0.545
	Innomed 2	174	168 (127–233)	4.25	0.550	171	163 (128–237)	1.12	0.888	4.3	0.831
	Clot activator tube	167	161 (127–220)			169	165 (130–230)				0.835

Note: IQR—interquartile range, *p* < 0.05 was considered statistically significant, * Centrifuged at 3500 rpm.

**Table 5 diagnostics-15-02398-t005:** Biochemical test validation of Innomed 1 and Innomed 2 tubes in a multicenter study.

**Parameter**	**Clinical Significance Limits**		**Healthy**						**Patient**					
			**Bias (%)**	**Heparin**		**Innomed 1**		** *p* ** **-Value**	**Bias (%)**	**Heparin**		**Innomed 1**		** *p* ** **-Value**
	**JHH** **(%)**	**Bias_d_ (%)**		**Range**	**Mean (SEM)**	**Range**	**Mean (SEM)**			**Range**	**Mean (SEM)**	**Range**	**Mean (SEM)**	
GLU (mmol/L)	8	2.34	5.32	3.17–6.22	5.01 (0.20)	4.61–6.17	5.28 (0.11)	0.033	2.93	3.83–23.94	7.41 (1.34)	4.24–24.06	7.62 (1.25)	0.945
ALB (g/L)	7	1.43	0	33.6–48.0	43.0 (1.4)	33.7–48.0	43.0 (1.6)	0.841	−2.33	28.1–77.2	43.0 (3.0)	28.4–48.4	42.0 (1.9)	0.538
ALP (U/L)	10	6.72	0.42	36–120	72.2 (6.11)	36–119	72.5 (6.16)	0.752	3.17	0.39–117	72.5 (7.30)	0.42–116	74.8 (7.70)	0.906
ALT (U/L)	15	11.48	3.75	8–127	26.7 (8.73)	8–127	27.7 (8.24)	0.980	8.49	13–70	31.8 (6.06)	7–69.5	34.5 (6.10)	0.962
AST (U/L)	10	6.54	2.05	7.9–40	19.5 (2.94)	6–42	19.9 (2.84)	0.864	6.9	13–189	31.9 (10.83)	13–185	34.1 (10.09)	0.966
BIL-D (µmol/L)	NA	14.20	0	1.03–6.16	3.42 (0.34)	0.86–6.67	3.42 (0.51)	0.903	0	0.00–10.60	3.42 (0.68)	0.00–10.26	3.42 (0.68)	0.974
BIL-T (µmol/L)	10	8.95	0	3.08–14.36	8.55 (1.03)	3.42–13.85	8.55 (0.86)	0.879	0	1.03–17.44	8.55 (1.20)	1.03–17.10	8.55 (1.20)	0.827
BUN (mmol/L)	10	5.57	1.65	2.29–6.75	4.32 (0.44)	2.29–6.71	4.39 (0.42)	0.982	5.56	2.14–29.27	6.43 (1.76)	2.14–28.90	6.78 (1.67)	0.992
CHOL (mmol/L)	5	4.10	−1.21	4.33–7.95	5.57 (0.30)	4.20–7.70	5.50 (0.27)	0.764	−1.18	2.44–7.28	5.26 (0.42)	2.38–7.36	5.19 (0.39)	0.823
CRE (µmol/L)	7	3.96	0	49.50–95.47	70.72 (4.42)	48.62–90.29	7.07 (4.42)	0.700	10	47.74–1249.06	176.80 (109.62)	45.97–1269.08	194.48 (103.43)	0.982
HDL (mmol/L)	10	5.61	−0.71	0.91–2.49	1.46 (0.12)	0.80–2.38	1.45 (0.12)	0.740	−6.21	0.47–3.16	1.33 (0.16)	0.47–2.95	1.25 (0.16)	0.868
LDH (U/L)	10	4.30	−1.9	111–236	16.31 (8.69)	108–238	160 (8.91)	0.668	−0.68	129–371	191.9 (18.61)	128–373	190.6 (18.93)	0.928
LDL (mmol/L)	10	5.46	−0.81	2.36–5.34	3.51 (0.27)	2.28–5.16	3.48 (0.25)	0.834	0.61	0.85–10.36	3.40 (0.59)	0.83–10.36	3.42 (0.55)	0.922
TP (g/L)	5	1.36	−1.3	63.0–86.0	77.0 (1.8)	64.0–87.0	76.0 (1.7)	0.669	−1.32	65.0–86.0	76.0 (1.6)	66.0–85.0	75.0 (1.5)	0.625
TRIG (mmol/L)	5	9.57	−1.43	0.51–2.54	1.43 (0.18)	0.44–2.42	1.41 (0.17)	0.683	4.46	0.47–6.84	2.14 (0.49)	0.46–6.76	2.24 (0.46)	0.874
UA (µmol/L)	10	4.87	−1.89	172.49–528.37	315.24 (27.95)	154.65–522.42	308.30 (29.14)	0.925	1.67	160.60–630.49	356.88 (38.66)	172.49–618.59	362.83 (36.88)	0.907
Na^+^ (mmol/L)	3	0.23	2.36	135–145.2	139.8 (0.91)	137–149.6	143.1 (1.20)	0.000	2.36	133.2–147.3	140 (1.15)	136–152.7	143.3 (1.35)	0.002
K^+^ (mmol/L)	8	1.81	5.26	3.28–4.41	3.8 (0.08)	3.36–6.76	4 (0.18)	0.263	0	3.28–5.81	4 (0.18)	3.31–5.85	4 (0.17)	0.802
Cl^−^ (mmol/L)	5	0.50	2.87	100.6–110	104.5 (0.77)	102.9–113	107.5 (0.92)	0.000	2.79	95.8–110.8	104 (3.88)	97.5–115.3	106.9 (1.44)	0.012
HCO_3_^−^ (mmol/L)	15	1.70	1.19	20.9–29	25.2 (0.68)	20–30	25.5 (0.76)	0.535	2.01	16.2–29.2	24.9 (1.27)	16–31	25.4 (1.35)	0.590
TSH (mIU/L)	NA	14.14	0.00	660–4300	2200 (410)	640–4300	2200 (410)	0.983	−3.26	2–46,030	9200 (4510)	3–15,820	8900 (4460)	0.969
FT3 (pmol/L)	NA	4.80	3.30	3.23–5.25	4.61 (0.18)	3.09–5.21	4.46 (0.26)	0.847	0.0	3.16–15.97	5.22 (1.29)	3.23–1.54	5.22 (1.23)	0.999
FT4 (pmol/L)	NA	2.98	0.00	9.52–19.44	12.87 (1.03)	9.27–18.28	12.87 (2.19)	0.861	0.0	4.12–22.52	11.58 (1.93)	3.47–22.38	11.58 (1.93)	0.789
T3 (nmol/L)	NA	5.19	1.08	1.07–2.03	1.43 (0.08)	1.10–2.02	1.41 (0.08)	0.898	−0.55	0.28–3.32	1.16 (0.26)	0.27–3.28	1.15 (0.26)	0.987
**Parameter**	**Clinical Significance Limits**		**Healthy**						**Patient**					
			**Bias (%)**	**Clot Activator**		**Innomed 2**		** *p* ** **-Value**	**Bias (%)**	**Clot activator**		**Innomed 2**		** *p* ** **-Value**
	**JHH** **(%)**	**Bias_d_ (%)**		**Range**	**Mean (SEM)**	**Range**	**Mean (SEM)**			**Range**	**Mean (SEM)**	**Range**	**Mean (SEM)**	
GLU (mmol/L)	8	2.34	5.98	4.29–5.87	3.04 (0.16)	4.78–5.94	5.21 (0.11)	0.149	3.00	3.82–23.89	7.21 (1.90)	4.24–24.05	7.41 (1.89)	0.937
ALB (g/L)	7	1.43	1.76	41.0–47.0	44.7 (0.5)	40.9–46.5	44.7 (0.5)	0.283	−0.69	36.0–49.0	43.2 (1.4)	35.9–48.4	42.9 (1.3)	0.444
ALP (U/L)	10	6.72	0.93	58.0–105.0	126.4 (8.65)	57.0–103.0	74.9 (4.77)	0.918	−0.59	54.0–117.0	84.5 (6.14)	54.0–116.0	84.0 (6.06)	0.954
ALT (U/L)	15	11.48	0.00	9.8–49.2	7.63 (0.08)	9.7–51.3	22.8 (4.52)	0.992	−1.72	22.0–70.0	40.8 (5.50)	21.3–69.5	40.1 (5.47)	0.927
AST (U/L)	10	6.54	2.00	6.5–33.2	125.6 (15.14)	6.6–35.1	15.3 (2.48)	0.938	−2.26	17.7–46.4	26.5 (2.87)	16.2–47.9	25.9 (3.10)	0.894
BIL-D (µmol/L)	NA	14.20	5.00	1.54–6.16	3.25 (0.51)	1.71–6.16	3.25 (0.51)	0.775	0.00	2.74–10.60	4.28 (0.68)	2.74–10.26	4.28 (0.68)	0.974
BIL-T (µmol/L)	10	8.95	0.00	3.25–14.02	8.55 (1.20)	3.25–13.85	8.55 (1.20)	1.000	−1.79	6.33–17.44	9.58 (1.03)	5.81–17.10	9.41 (1.03)	0.956
BUN (mmol/L)	10	5.57	0.00	2.36–6.75	4.64 (0.47)	2.28–6.78	4.64 (0.47)	0.975	0.00	3.43–19.88	7.96 (1.73)	3.43–19.71	7.96 (1.72)	0.998
CHOL (mmol/L)	5	4.10	1.70	4.46–6.67	5.09 (0.25)	4.40–6.53	5.09 (0.25)	0.809	−0.53	3.29–6.53	4.90 (0.37)	3.35–6.44	4.88 (0.36)	0.959
CRE (µmol/L)	7	3.96	4.11	52.16–84.86	61.88 (3.35)	48.62–87.52	61.88 (3.37)	0.639	0.86	60.11–1249.06	306.35 (137.22)	60.11–1271.08	309.40 (139.55)	0.990
HDL (mmol/L)	10	5.61	2.01	1.15–1.90	1.42 (0.08)	1.06–1.85	1.39 (0.08)	0.802	0.67	0.02–0.37	0.09 (0.07)	0.81–1.51	1.14 (0.07)	0.941
LDH (U/L)	10	4.30	1.16	99–204	156 (8.81)	108–210	157 (10.13)	0.895	−1.15	31.5–58.5	44.5 (20.84)	141–373	206 (21.70)	0.937
LDL (mmol/L)	10	5.46	−0.95	2.53–4.58	3.27 (0.22)	2.49–4.48	3.24 (0.22)	0.922	−1.53	3.70–9.61	5.41 (0.35)	1.58–4.91	3.17 (0.34)	0.921
TRIG (mmol/L)	5	9.57	−1.04	0.64–2.41	1.42 (0.17)	0.62–2.37	1.40 (0.17)	0.953	−0.26	0.08–0.10	0.09 (0.27)	0.46–3.08	1.75 (0.27)	0.991
UA (µmol/L)	10	4.87	−0.98	291.45–522.42	362.83 (21.95)	285.50–522.42	359.28 (21.95)	0.909	−1.46	2461.57–16,242.19	9228.35 (39.26)	249.82–618.59	401.48 (38.08)	0.914
Na^+^ (mmol/L)	3	0.23	1.58	136.4–141.8	139.30 (0.58)	139.0–143.6	141.50 (0.52)	0.009	1.94	4.3–10.6	6.85 (0.56)	137.9–144.0	141.80 (0.66)	0.005
K^+^ (mmol/L)	8	1.81	1.69	3.8–4.6	4.13 (0.06)	3.8–4.7	4.20 (0.08)	0.481	0.73	135.2–141.0	139.10 (0.18)	3.3–5.2	4.15 (0.17)	0.919
Cl^−^ (mmol/L)	5	0.50	1.85	99.9–104.0	102.70 (0.41)	103.2–106.7	104.60 (0.38)	0.003	1.66	95.8–108.0	102.20 (1.18)	97.5–109.4	103.90 (1.07)	0.319
HCO_3_^−^ (mmol/L)	15	1.70	2.92	21.0–26.5	24.00 (0.58)	21.4–27.3	24.70 (0.62)	0.421	−0.37	24.1–29.2	26.80 (0.52)	24.4–28.4	26.70 (0.49)	0.912
TSH (mIU/L)	NA	14.14	0.00	690–2370	1300 (180)	670–2330	1300 (170)	0.924	0.00	910–3140	1500 (210)	930–3140	1500 (210)	0.500
FT3 (pmol/L)	NA	4.80	2.60	4.04–5.68	4.91 (0.18)	4.16–5.56	4.91 (0.15)	0.864	0.00	4.17–5.70	5.07 (0.18)	4.19–5.85	5.07 (0.22)	0.678
FT4 (pmol/L)	NA	2.98	0.00	90.85–150.09	123.69 (6.05	90.39–168.72	126.86 (7.47)	0.736	2.98	87.30–147.70	122.70 (6.18)	80.49–157.19	128.70 (7.85)	0.555

Note: Bias_d_ = desirable quality specifications for bias, SEM = standard error of the mean, Bias = difference between the compared tube results, GLU = Glucose, BUN = Blood urea nitrogen, CRE = Creatinine, CHOL = Total cholesterol, TRIG = Triglyceride, HDL = High-density lipoprotein, LDL = Low-density lipoprotein, UA = Uric acid, AST = Aspartate aminotransferase, ALT = Alanine aminotransferase, ALP = Alkaline phosphatase, BIL-D = Direct bilirubin, BIL-T = Total bilirubin, ALB = Albumin, LDH = lactate, Na^+^ = Sodium, K^+^ = potassium, Cl^−^ = Chloride, HCO_3_^−^ = Total carbon dioxide, TSH = Thyroid stimulating hormone, FT3 = Triiodothyronine, free, FT4 = Thyroxine, free, T3= Triiodothyronine dehydrogenase, %Bias: (Mean test − Mean target)/Mean target × 100, %Bias_d_: Bias desirable quality specifications for bias, %JHH: Percent changes in analyte (Johns Hopkins Hospital Laboratory Acceptability Thresholds) Acceptable criteria and significant limits of clinical limits and/or statistical limits (*p* > 0.05). NA: Not applicable.

## Data Availability

The data presented in this study are available on request from the corresponding author due to privacy, legal or ethical restrictions.

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
