# Peer review of "Novel Blood Collection Tubes Improve Sample Preservation in a Multicenter Study in Thailand"

_diagnostics, 2025, doi:10.3390/diagnostics15182398_

Round 1

Reviewer 1 Report

Comments and Suggestions for Authors

Dear Authors,

This is an interesting manuscript, but it needs to be improved. Below I listed my comments:

In the methods of the abstract: Why did you decide on these two tubes, much more often are tubes with gel, without anticoagulants or with thrombin for biochemistry, or tubes with EDTA for hematology and blood picture parameters, or with citrate for coagulation parameters.

also specify the time period in which the samples are centrifuged

In the Conclusion of the abstract: here you mention diabetes, but nowhere in the objective or introduction is it stated that it is therefore of interest to examine these test tubes and what NCD is, explain the abbreviation. From the methods, it is concluded that it is of interest to examine the insensitivity of test tubes to hemolysis and interference, but not stability, it should be written more clearly what the aim of the study is and why this research is being done, as well as what the results are and what the conclusion is

In the Introduction "Managing Emergencies..." this should be corrected, perhaps stating, practicality, thriftiness, patient care because blood is taken in one step, not in multiple tubes... what is the benefit

In the third paragraph, verification/validation, you should have a literary source for these statements, list the references at the end of the sentence

cite the GP34A standard at the end of the sentence which is a literary source.

At the end of the data analysis: was this sentence put here by mistake? and since the determination from two test tubes can be considered as a measurement by two methods, I think it would be important to examine the Passing-Bablock test and the agreement of the methods for each parameter, which automatically obtains the value of the bias, and even whether the deviation is systematic or proportional

In Tables: I think all units should be traceable and written according to the SI system of units e.g LDL in mmol/L

In the Discussion: here you now start with glucose stability as an important item of this paper, and in the paper you examined a large number of different parameters. Is glucose stability important as a clinical data or as an interference with other tests, why is it important in general biochemical determinations

In conclusion: what the wellness program means in health care is not clear

Author Response

Reviewer 1

Comment 1: In the methods of the abstract: Why did you decide on these two tubes, much more often are tubes with gel, without anticoagulants or with thrombin for biochemistry, or tubes with EDTA for hematology and blood picture parameters, or with citrate for coagulation parameters. also specify the time period in which the samples are centrifuged
Response1: The rationale for selecting these two tubes is as follows: These two tubes were selected because they were newly developed to allow multi-parameter testing (biochemistry, hematology, HbA1c, thyroid hormones, and tumor markers) from a single blood sample.

We have added the requested details to the Abstract-Methods section. (On page 1, lines 26-27)

“Subsequently, samples were centrifuged at 3,500 rpm for 10 min and analyzed immediately after separation, not exceeding 2 h from collection.”

Comment 2: In the Conclusion of the abstract: here you mention diabetes, but nowhere in the objective or introduction is it stated that it is therefore of interest to examine these test tubes and what NCD is, explain the abbreviation. From the methods, it is concluded that it is of interest to examine the insensitivity of test tubes to hemolysis and interference, but not stability, it should be written more clearly what the aim of the study is and why this research is being done, as well as what the results are and what the conclusion
Response2:
We have added the requested details to the Abstract Purpose section. (On page 1, lines 20-21)

aiming to assess their performance in glucose stability, hemolysis resistance, and biochemical interferences relevant to diabetes and non-communicable disease (NCD) monitoring.

Abstract Conclusion section. (On page 1, lines 36-37)

thereby emphasizing their clinical relevance in chronic disease management.

Comment 3: In the Introduction "Managing Emergencies..." this should be corrected, perhaps stating, practicality, thriftiness, patient care because blood is taken in one step, not in multiple tubes... what is the benefit

Response3: We have added the requested details to the Introduction section. (On page 2, lines 49-51)

“using a single blood collection tube offers practicality, thriftiness, and improved patient care, as blood is collected in one step rather than multiple tubes, thereby reducing patient discomfort and pre-analytical variability”

Comment 4: In the third paragraph, verification/validation, you should have a literary source for these statements, list the references at the end of the sentence cite the GP34A standard at the end of the sentence which is a literary source.

Response4: We have added the CLSI GP34A guideline as reference number 15 in the Introduction (third paragraph) to support the statements regarding verification and validation.

Comment 5: At the end of the data analysis: was this sentence put here by mistake? and since the determination from two test tubes can be considered as a measurement by two methods, I think it would be important to examine the Passing-Bablok test and the agreement of the methods for each parameter, which automatically obtains the value of the bias, and even whether the deviation is systematic or proportional

Response5: We thank the reviewer for the suggestion. In this study, we prioritized the Johns Hopkins Hospital (JHH) criteria as the main clinical limits, and applied the desirable quality specifications for bias (Biasd) when JHH thresholds were not available. As our study followed CLSI GP34A for verification/validation of blood collection tubes, we focused on meeting predefined clinical and analytical criteria rather than performing method-comparison regression such as Passing–Bablok.

Comment 6: In Tables: I think all units should be traceable and written according to the SI system of units e.g LDL in mmol/L

Response6: We thank the reviewer for the comment. We have revised all tables to ensure that all measurement units are expressed in SI units.

Comment 7: In the Discussion: here you now start with glucose stability as an important item of this paper, and in the paper you examined a large number of different parameters. Is glucose stability important as a clinical data or as an interference with other tests, why is it important in general biochemical determinations

Response7: We appreciate the reviewer’s comment. Glucose stability was highlighted because, in terms of clinical data, accurate glucose measurement is essential for the diagnosis and monitoring of diabetes. In addition, glycolysis can influence other biochemical parameters, thereby causing interferences in general biochemical determinations.

Comment 8: In conclusion: what the wellness program means in health care is not clear

Response 8: We have added the requested details to the conclusion section.  (On page 15, lines368-371)

In this study, the wellness program is preventive health services and NCD monitoring supported by reliable laboratory testing with Innomed tubes. Such integration may enhance preventive care and chronic disease management within healthcare systems.

Reviewer 2 Report

Comments and Suggestions for Authors

The paper "Novel Blood Collection Tubes Improve Sample Preservation in a Multicenter Study in Thailand" is aimed at optimizing the pre-analytical phase of laboratory diagnostics through the implementation of new blood collection tubes, Innomed 1 and 2. These tubes have the potential to significantly streamline laboratory diagnostics by enabling multi-parameter testing from a single sample. The results obtained demonstrate high reliability and clinical acceptability of these tubes for various types of laboratory tests. However, several aspects should be addressed and revised:

  1. What is the clinical relevance of the study? I recommend that the authors more clearly emphasise the practical significance of the study in the introduction.
  2. In the Materials and Methods section, the authors should provide more detailed donor information, including gender and age.
  3. Did the authors conduct a power analysis of the study? If possible, please add a power analysis to demonstrate whether the sample size is sufficient to detect significant differences.
  4. Was any correction for multiple comparisons applied (e.g., Bonferroni correction)?
  5. Why is the glucose stability study limited to 8 hours? In some laboratory settings, samples may remain in tubes for up to 24 hours. Additional data or clarification on this limitation should be included.
  6. Line 111 and onwards - check that all the names of the probors are country specific.
  7. Table 1 - The authors say there was a healthy group (n=12) and patient group (n=24). But further in the table it is not clear how many donors there were for Heparin and how many for Innomed 1.
  8. All parameters to be measured must be listed in the methodology.
  9. In Table 2, am I correct in understanding that 103 is the same as 103. If so, make sure that degree 10 is at the top.
  1. Please include some perspectives on future research. As I am confused by the fact that the authors have not conducted clinical trials.

Author Response

Comment 1: What is the clinical relevance of the study? I recommend that the authors more clearly emphasise the practical significance of the study in the introduction.

Response1: We have added the requested details to the Materials and Introduction section.(On page 2, lines 81-85)  The clinical relevance of this study is underscored by the capacity of Innomed tubes to preserve glucose stability within clinically acceptable limits for up to 8 hours and to resist hemolysis under different processing conditions, thereby supporting accurate biochemical testing and reliable monitoring of diabetes and other non-communicable diseases.

Comment 2: In the Materials and Methods section, the authors should provide more detailed donor information, including gender and age.
Response2:
We have added the requested details to the Materials and Materials and Methods section. 

(On page 3, lines108-109/111-118)

Innomed 1 and 2 tubes were used to collect venous blood samples from 36 participants, comprising 12 healthy individuals and 24 patients. Healthy participants were aged ≥20 years, free from disease, had no recent injuries or blood loss, and laboratory test results within reference ranges. The patient group included 8 with diabetes mellitus, 8 with hypercholesterolemia, and 8 with multiple chronic non-communicable diseases (e.g., diabetes combined with hypercholesterolemia or hypertension), also aged ≥20 years and without recent injuries or blood loss. Both groups had equal numbers of male and female participants. Exclusion criteria included pregnancy, anemia, cancer, hepatitis B or C infection, immunodeficiency (HIV), syphilis, or inability to provide sufficient blood volume

Comment 3: Did the authors conduct a power analysis of the study? If possible, please add a power analysis to demonstrate whether the sample size is sufficient to detect significant differences.

Response3: In this study, we did not perform a formal power analysis. The sample size was determined according to CLSI GP34A guidelines, which require sufficient participants to cover the analyte range, variability, and preanalytical conditions. Thus, 36 participants (12 healthy and 24 patients) were included, which we considered appropriate for this validation study.

Comment 4: Was any correction for multiple comparisons applied (e.g., Bonferroni correction)?

Response4: In this study, we did not apply a formal correction for multiple comparisons. This is because the study was primarily designed as a method validation and verification according to CLSI GP34A guidelines, where the focus was on assessing agreement between tubes rather than performing exploratory hypothesis testing across a large number of independent variables. Although multiple biochemical parameters were evaluated, the comparisons were structured and predefined, and the results were interpreted in the context of method comparison rather than discovery testing. Therefore, we considered a correction for multiple comparisons unnecessary in this setting.

Comment 5: Why is the glucose stability study limited to 8 hours? In some laboratory settings, samples may remain in tubes for up to 24 hours. Additional data or clarification on this limitation should be included.

Response5: In this study, glucose stability was evaluated for up to 8 hours, which corresponds to the typical preanalytical timeframe in routine hospital practice, where samples are usually processed within the same working day. This design is also consistent with the CLSI GP34A guideline, which emphasizes verification under representative preanalytical conditions. We acknowledge that in some laboratory settings samples may be stored for longer periods (e.g., up to 24 hours), and additional studies covering extended storage conditions would be valuable for future research.

we have revised the Discussion section (page 13, lines 278–282) by adding clarification on the rationale for the 8-hour limit “In this study, glucose stability was assessed for up to 8 hours, reflecting the routine preanalytical timeframe in hospital laboratories and consistent with CLSI GP34A recommendations. We acknowledge that in some settings samples may be stored long-er (e.g., 24 hours), and further studies under extended conditions would be valuable

Comment 6: Line 111 and onwards - check that all the names of the probors are country specific.

Response6: We have corrected the information accordingly in the Materials and Methods section (page 3 lines 120-121)

Comment 7: Table 1 - The authors say there was a healthy group (n=12) and patient group (n=24). But further in the table it is not clear how many donors there were for Heparin and how many for Innomed 1.

Response7: We have provided additional explanation in the Materials and Methods section.

(page 3, lines 109-111)
“For each donor, venous blood was collected in parallel into a commercial tubes (con-trol) and an Innomed tube (test), enabling direct comparison between tube types.”

Comment 8: All parameters to be measured must be listed in the methodology.

Response 8: We thank the reviewer for this helpful comment. We have added the full list of measured parameters in the Methods (Analytical methods section) to provide clarity regarding all biochemical, hematological, endocrine, and tumor marker tests included in this study. (On page 3, lines 122-128)

The following parameters were measured: glucose; 19 routine biochemical analytes (albumin, ALP, ALT, AST, direct bilirubin, total bilirubin, BUN, cholesterol, creatinine, HDL, LDH, LDL, total protein, triglycerides, uric acid, sodium, potassium, chloride, bicarbonate); HbA1c; thyroid hormones (TSH, FT3, FT4, T3); tumor markers (CEA, PSA); and complete blood count parameters (hemoglobin, hematocrit, RBC, WBC, platelet count, MCV, MCH, MCHC, RDW, and differential counts of neutrophils, lymphocytes, monocytes, eosinophils, and basophils).

Comment 9: In Table 2, am I correct in understanding that 103 is the same as 103. If so, make sure that degree 10 is at the top.

Response 9: We have corrected the notation in Table 2  to use superscript format  according to SI unit standards.

Comment 10: Please include some perspectives on future research. As I am confused by the fact that the authors have not conducted clinical trials.

Response 10: We thank the reviewer for the comment. This study already included clinical trials and multicenter evaluations in several hospitals in Thailand. We have revised the Conclusion to note that future research should expand to larger populations and diverse healthcare systems to strengthen generalizability.

We have provided additional explanation in the Conclusion section. (page 15, lines 375-378)

This study encompassed clinical and multicenter evaluations in Thailand, confirming the applicability of Innomed tubes. Further investigations in larger and more diverse populations and healthcare systems are warranted to enhance generalizability.

Round 2

Reviewer 2 Report

Comments and Suggestions for Authors

The authors answered all questions.